# Psychological mechanisms of offset analgesia: The effect of expectancy manipulation

**Tibor M. Szikszay** [1]*, **Waclaw M. Adamczyk**[1,2], **Janina Panskus**[1], **Lotte Heimes**[1], **Carolin David**[1], **Philip Gouverneur**[3], **Kerstin Luedtke**[1]

**1** Institute of Health Sciences, Department of Physiotherapy, Pain and Exercise Research Luebeck (P.E.R. L.), Universität zu Lübeck, Luebeck, Germany, **2** Laboratory of Pain Research, Institute of Physiotherapy and Health Sciences, The Jerzy Kukuczka Academy of Physical Education, Katowice, Poland, **3** Institute of Medical Informatics, University of Luebeck, Luebeck, Germany

☯ These authors contributed equally to this work.
* tibor.szikszay@uni-luebeck.de

**Data Availability Statement:** Data is available in the supporting information.

**Funding:** The study was conducted thanks to support from the National Science Centre in Poland

## Abstract

A frequently used paradigm to quantify endogenous pain modulation is offset analgesia, which is defined as a disproportionate large reduction in pain following a small decrease in a heat stimulus. The aim of this study was to determine whether suggestion influences the magnitude of offset analgesia in healthy participants. A total of 97 participants were randomized into three groups (hypoalgesic group, hyperalgesic group, control group). All participants received four heat stimuli (two constant trials and two offset trials) to the ventral, non-dominant forearm while they were asked to rate their perceived pain using a computerized visual analogue scale. In addition, electrodermal activity was measured during each heat stimulus. Participants in both intervention groups were given a visual and verbal suggestion about the expected pain response in an hypoalgesic and hyperalgesic manner. The control group received no suggestion. In all groups, significant offset analgesia was provoked, indicated by reduced pain ratings ($p < 0.001$) and enhanced electrodermal activity level ($p < 0.01$). A significant group difference in the magnitude of offset analgesia was found between the three groups ($F_{[2,94]} = 4.81$, $p < 0.05$). Participants in the hyperalgesic group perceived significantly more pain than the hypoalgesic group ($p = 0.031$) and the control group ($p < 0.05$). However, the electrodermal activity data did not replicate this trend ($p > 0.05$). The results of this study indicate that suggestion can be effective to reduce but not increase endogenous pain modulation quantified by offset analgesia in healthy participants.

## Introduction

Endogenous pain modulation has been proposed and is discussed as a leading feature of the nociceptive system that can promote or protect the individual against the transition from acute to chronic pain [1, 2]. In general, pain modulation can be assessed through experimental paradigms which are believed to reflect complex inhibitory and faciliatory mechanisms within the neuroaxis [3]. Thus, within the peripheral and central nervous system, a variety of individual transmitters and specific receptor types are involved in the modulation and expression of

(2020/04/X/HS6/01927). The funders had no role in study design, data collection and analysis, decision to publish, or preparation of the manuscript.

**Competing interests:** The authors have declared that no competing interests exist.

descending inhibition and facilitation [4]. In the central nervous system, pain can be modulated by cognitive, affective and motivational factors [5] including beliefs and expectations [6]. Furthermore, the efficiency of these modulatory pathways can be assessed in the laboratory setting, using paradigms such as conditioned pain modulation (CPM: the "pain inhibits pain") [2] and/or offset analgesia (OA).

Offset analgesia can be defined as a disproportionally large pain decrease after a minor noxious stimulus intensity reduction [7]. This test procedure is discussed to indicate the efficiency of the descending inhibitory pain modulation system in humans [8]. For almost 20 years, numerous studies have attempted to identify the processes underlying OA, but the physiological mechanisms have not yet been fully understood. Both peripheral [9–11], spinal [12] and supra-spinal mechanisms [9, 13–17] have been shown to contribute to OA. However, few experimental procedures in the past effectively modulated the OA effect. For instance, the modulatory influence of primary afferents [11, 18] or secondary noxious stimuli [19] are exceptions rather than common findings expanding our knowledge of OA. In contrast, all pharmacological attempts failed to affect OA [20].

Interestingly, psychological interventions have never been used in the context of OA. It is of clinical interest, whether psychological processes influence endogenous pain modulation, especially since—amongst others- hypo- or hyperalgesic suggestion have been shown to effectively decrease [21] or increase [22] pain perception, respectively. For example, it has been demonstrated that by administering a hyperalgesic suggestion prior the application of a noxious stimulus, healthy subjects felt more pain [23]. The putative mechanism of such an intervention relates to expectations [24], which has already been observed in CPM experiments [25, 26] but not yet in OA.

This experiment attempted to influence the magnitude of OA by manipulating participants' expectations using suggestions. In this study, suggestions were used to modulate OA selectively, that is, by changing the pain response in the final temperature phase of the paradigm. Therefore, the á priori hypothesis implied that suggestion would influence the OA effect in a bidirectional manner, i.e., analgesia was expected to be increased or decreased, respectively, compared to a control group that was not exposed to any form of suggestion.

## Materials and methods

### Study design

This experimental study was conducted as a randomized controlled trial in which healthy, pain-free participants were randomly divided (counterbalanced) into two intervention groups and one control group. Both intervention groups received either a hypoalgesic or a hyperalgesic suggestion related to the pattern of the subsequently applied heat pain within an OA paradigm. The control group received no suggestion. All participants received the identical information about the exact temperature course of the heat stimuli beforehand. In order to perform the suggestions as authentically as possible, a cover story was told to all participants at the beginning of the study. All participants were blinded to the true purpose of the study. The study was previously approved by the ethics committee of the University of Lübeck (file number: 21–028) and pre-registered in the Open Science Framework (https://osf.io/69eyp). All participants provided oral and written informed consent. An overview of the study design is provided in Fig 1.

### Study population

Healthy, pain-free participants aged 18 to 65 years were recruited on the campus of the University of Lübeck. All participants had to subjectively confirm that they were healthy.

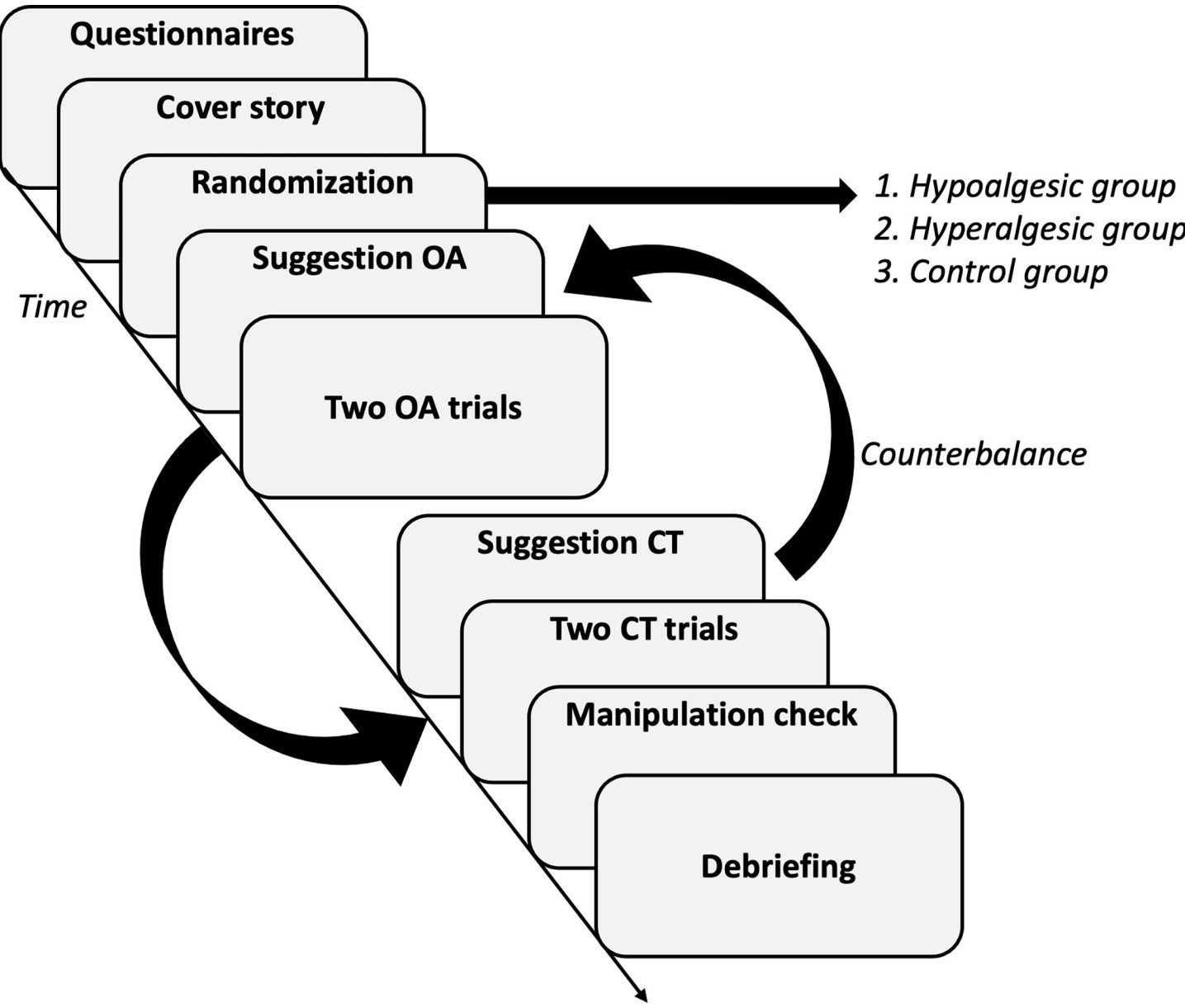

**Fig 1. Study design.** Before randomization, participants were instructed and a cover story was provided. Participants were told that in this study, changes in electrodermal activity (EDA) would be assessed as a measure of the autonomic nervous system during experimental heat stimuli and used to predict the perception of pain (cover story). Participants were assigned to either i) the hypoalgesic group with suggestion towards profound hypoalgesia following the temperature reduction, ii) the hyperalgesic group with verbal suggestion towards hyperalgesia following the temperature reduction (see the example above), or iii) the control group with no intervention. Regardless of the group assignment participants were exposed to two offset and two constant trials provided in a counterbalanced manner.

Furthermore, participant had no cardiovascular, systemic, psychiatric or neurological disease. Furthermore, all participants were excluded if they had a history of chronic pain (> 3 months) within the last 2 years or had experienced pain (including headache, toothache, muscle soreness, etc.) within the last week prior to study participation. In addition, the use of medication, excluding contraceptives, in the last 48 hours was an exclusion criterion. Furthermore, participants were asked not to drink alcohol, exercise, or take pain medication for 24 hours prior to participation in the study and not to drink coffee or smoke cigarettes for 4 hours prior to study participation.

## Equipment

A Pathway CHEPS (Contact Heat -Evoked Potential Stimulator) with a contact area of 27mm diameter was used for the application of the heat stimuli (Medoc, Ramat Yishai, Israel). The thermode was attached to the non-dominant volar forearm approximately 10 cm below the elbow using a blood pressure cuff with a pressure of 25 mmHg. A computerized visual analog scale hardware device (COVAS; with the range 0 = "no pain" to 100 = "most tolerable pain") was used for continuous assessment of pain intensity (Medoc, Ramat Yishai, Israel). Furthermore, during heat stimuli, electrodermal activity (EDA; respiBAN Professional, Plux, Lisbon, Portugal) was measured using two Ag/AgCl hydrogel electrodes (Covidien / Kendall, Dublin, Ireland) at the medial phalanx of index and middle finger of the non-dominant arm (sampling rate of 1000 Hz, PLUX Wireless Biosignals, S.A., Portugal). Electrodermal activity depends on sweat secretion, which is closely related to autonomic nervous system activity [27]. EDA was used to test, if verbal suggestion influences physiological responses and if EDA can be used as an objective marker for OA.

## Experimental heat stimulation

Two constant trials (CT) and two offset trials (OT) were performed on the non-dominant volar forearm, so that the participants were presented with a total of four heat stimuli. The order in which trials were presented was randomized in a counterbalanced fashion. A two-minute pause was kept between each stimulus, during which the thermode was moved on the forearm by a few centimeters. The temperature's rise and fall rate for all heat stimuli was 15°C/second. During CT, the temperature increased from a baseline level of 35°C to 46°C and remained constant for 40 seconds before returning to the baseline level. During OT, the temperature first increased to 46°C (T1) for 10 seconds, then increased to 47°C (T2) for 10 seconds, and finally decreased again to 46°C (T3) lasting 20 seconds. The temperature pattern of the two trials can be seen in Fig 1. These figures were shown to the participants before the application of the heat stimuli. During the application of the heat stimuli, participants were asked to rate perceived pain continuously and as precisely using the COVAS.

## Suggestion

The participants were provided with a cover story. It was explained that the aim of the study was to find out whether the subjective sensation of pain could be "read out" from the physiological reaction of the body (skin conductance) and thus be predicted. A cover story was necessary to also justify the introduction of suggestion as credibly as possible without participants becoming skeptical or biased.

The hypoalgesic or hyperalgesic groups received suggestions about the expected pain pattern and the pain intensity of the applied heat stimuli. The hypoalgesic group received a suggestion to enhance the effect of OA and adaptation to CT, i.e., to reduce pain perception. The hyperalgesic group, on the other hand, received a suggestion, which was intended to reduce the effect of OA and adaptation to CT, i.e., to increase pain perception. The expected pain pattern was manipulated verbally (S1 File) and supported with the graphical presentation of the assumed pain pattern (S1 Fig) and took place directly after the explanation of the temperature gradients, i.e. immediately before application of the respective heat stimulus.

## Questionnaires

Before starting the heat application, participants were asked to complete several questionnaires: The Patient Health Questionnaire (PHQ-9) includes nine questions about depression

[28]. While the Pain Vigilance and Awareness Questionnaire (PVAQ) measures pain perception and pain awareness [29], the Pain Sensitivity Questionnaire (PSQ) can be used to determine the subject's pain sensitivity [30]. The State-Trait Anxiety Inventory-SKD (STAI-SKD) was also collected to determine the participant's current state anxiety before the experiment [31]. The Social Desirability Scale-17 (SDS-17) was used to measure the participant's social desirability [32]. Furthermore, the Mindful Attention and Awareness Scale (MAAS) was used to measure dispositional mindfulness [33] and the Life-Orientation Test (LOT-R) was used to measure individual differences between optimism and pessimism based on personality traits [34].

## Manipulation check

To assess the effect of suggestion on pain perception during the OA paradigm, a manipulation check was performed immediately after pain assessment. The following was asked separately for OT and CT: "Please try to recall the moment immediately after receiving heat stimuli. Did you perceive the pain as in the previously displayed figures?" Participants provided a binary (yes vs. no) response.

## Statistical analysis

In the absence of studies investigating OA and verbal suggestion, a meta-analysis examining the effect of verbal suggestion on general pain perception was used to calculate the sample size [22]. With the lowest reported effect size of 0.66 (Cohen's d), a power of 80%, and an alpha of 0.05, a total number of 30 participants in each group (total = 90) was required (G*Power, University of Düsseldorf [35]) to demonstrate a significant difference between experimental and control groups.

COVAS data from the Medoc software and the EDA signals were synchronized. The time-series data were down-sampled to a frequency of 1 Hz by using the "resample" function of the python package "Pandas" (Python 3.9.7, pandas 1.4.2). Here, multiple data points are aggregated and replaced by their average. No further preprocessing steps were performed. All other statistical analyses were performed using the IBM Statistical Package for Social Science (SPSS version 26, Armonk, NY). The three groups were tested for group differences using age, BMI, sex, dominant hand, and questionnaire data. One-way analysis of variance (ANOVA), Kruskal-Wallis tests, or chi-square tests were used accordingly.

Differences between the groups in their initial pain response at the beginning of the heat stimuli were examined. For this purpose, pain ratings were averaged from the last 5 seconds of the T1 and T2 interval. Separately for OTs and CTs (mean value of the two CTs and two OTs) one-way ANOVAs were used to analyze differences between the groups. The primary outcome in this study was the magnitude of the pain response to the T3 interval (OT). To ensure that the magnitude of the OA effect was not under- or overestimated, the mean of 10 seconds centered in the T3 interval (secs. 25–34) were extracted. The 5 seconds at the beginning of the interval were not included, because the pain may still decrease during this time, and the 5 seconds at the end of the interval were not included, because the analgesic effect of OA usually decreases after approximately 15 seconds [36]. OT and CT (again mean of the two CTs and OTs) were analyzed separately, as both trials were also separately attempted to be influenced by suggestion. However, dependent t-tests were used to demonstrate whether the pain response and EDA signal from CT were significantly different from OT in each of the groups and thus whether there was an OA effect. To examine the effect of suggestion on OT and CT, a one-way ANOVA was conducted comparing the T3 interval pain response of the three groups as described. An additional method of analysis calculating the percentage difference of the

maximum pain ratings for T2 and the minimum pain rating for T3 is included in the support-
ing information (S2 File).

The EDA data were analyzed according to identical principles as the pain response. As part
of a secondary analysis of the EDA data in the control group, an ANOVA was performed with
the within factor "interval" (T1, T2, T3) and the within factor "trial" (OT, CT). Furthermore, a
dependent t-test was used to examine whether the magnitude of the EDA magnitude differed
within the temperature increase/decrease (i.e. comparing the EDA magnitude from the differ-
ence of T2 and T1 with the difference of T2 and T3).

If statistically significant main or interaction effects were detected, Bonferroni corrected
post-hoc t tests were conducted. The correlations between the pain response of the T3 (OT)
and the previously described questionnaires were calculated using the Spearman coefficient.
No data were missing at the time of analysis. A $p$ value of less than 0.05 was considered signifi-
cant for all comparisons.

## Results

A total of 97 participants (hypoalgesic n = 32, hyperalgesic n = 33, control group n = 32) were
included in this study. No significant differences were found between groups regarding base-
line characteristics (Table 1). The raw data are presented in the supporting information
(S3 File).

Mean pain curves and averaged pain from T3 intervals are presented in Figs 2 and 3, respec-
tively. No significant differences were found between all groups regarding the T1 interval for
either OT ($F_{(2, 94)}$ = 2.48, p = 0.09, $\eta^2_p$ = 0.06) or CT ($F_{[2, 94]}$ = 1.81, p = 0.17, $\eta^2_p$ = 0.04).
Dependent t-tests showed that OT regarding the T3 interval was significantly different from
CT in the pain ratings (hypoalgesic: $t_{[31]}$ = 6.3, p < 0.001, $d_z$ = 1.12; hyperalgesic: $t_{[32]}$ = 5.8,
p < 0.001, $d_z$ = 1.01; control: $t_{[31]}$ = 7.1, p < 0.001, $d_z$ = 1.25) as well as EDA (hypoalgesic: $t_{[31]}$
= 4.1, p < 0.001, $d_z$ = -0.71; hyperalgesic: $t_{[32]}$ = 4.5, p < 0.001, $d_z$ = -0.78; control: $t_{[31]}$ = 3.5,
p < 0.001, $d_z$ = -0.61) in all groups, indicating an OA effect within each of the groups.

**Table 1. Participant characteristics for each group.**

|  | Hypoalgesic (n = 32) | Hyperalgesic (n = 33) | Control (n = 32) | p |
|---|---|---|---|---|
| **Age $\bar{x}$ (SD)** | 29.7 (12.2) | 25.6 (9.0) | 28.7 (11.4) | 0.30 [a] |
| **BMI $\bar{x}$ (SD)** | 23.0 (2.3) | 23.1 (2.4) | 22.9 (3.2) | 0.97 [a] |
| **Female n (%)** | 17 (53.1) | 16 (48.5) | 20 (62.5) | 0.51 [b] |
| **Right-handed n (%)** | 30 (93.8) | 30 (90.9) | 27 (84.4) | 0.45 [b] |
| **PHQ9 M (IQR)** | 3.0 (1.0; 4.0) | 2 (2.0; 4.5) | 3 (1.3; 4.0) | 0.95 [c] |
| **PVAQ M (IQR)** | 41.5 (34.0; 49.8) | 38 (29.0; 46.6) | 36.5 (28.0; 48.0) | 0.32 [c] |
| **PSQ M (IQR)** | 3.2 (2.5; 4.4) | 3.5 (2.3; 4.2) | 2.8 (2.3; 3.7) | 0.44 [c] |
| **STAIT-SKD M (IQR)** | 6 (5.0; 7.0) | 6 (5.0; 7.0) | 6 (5.0; 7.0) | 0.76 [c] |
| **SDS-17 M (IQR)** | 12 (11.0; 14.0) | 12 (10.0; 13.5) | 10.5 (8.0; 13.0) | 0,08 [c] |
| **MAAS M (IQR)** | 65.0 (59.0; 72.0) | 66.0 (62.5; 71.5) | 66.0 (56.3; 72.8) | 0.72 [c] |
| **LOT-R M (IQR)** | 19.0 (16.3; 21,8) | 19.0 (17.0; 21.0) | 19.0 (16.3; 21.0) | 0.90 [c] |

$\bar{x}$: mean SD; Standard deviation; M: Median; IQR: Interquartile range; PHQ9: Patient Health Questionnaire; PVAQ: Pain Vigilance and Awareness Questionnaire; PSQ:
Pain Sensitivity Questionnaire; STAIT-SKD: State-Trait-Anxiety-Inventory-SKD; SDS-17: Social Desirability Scale-17; MAAS: Mindful Attention and Awareness Scale,
LOT-R: Life-Orientation-Test

[a] Analysis of variance

[b] Chi$^2$-Test

[c] Kruskal-Wallis-Test.

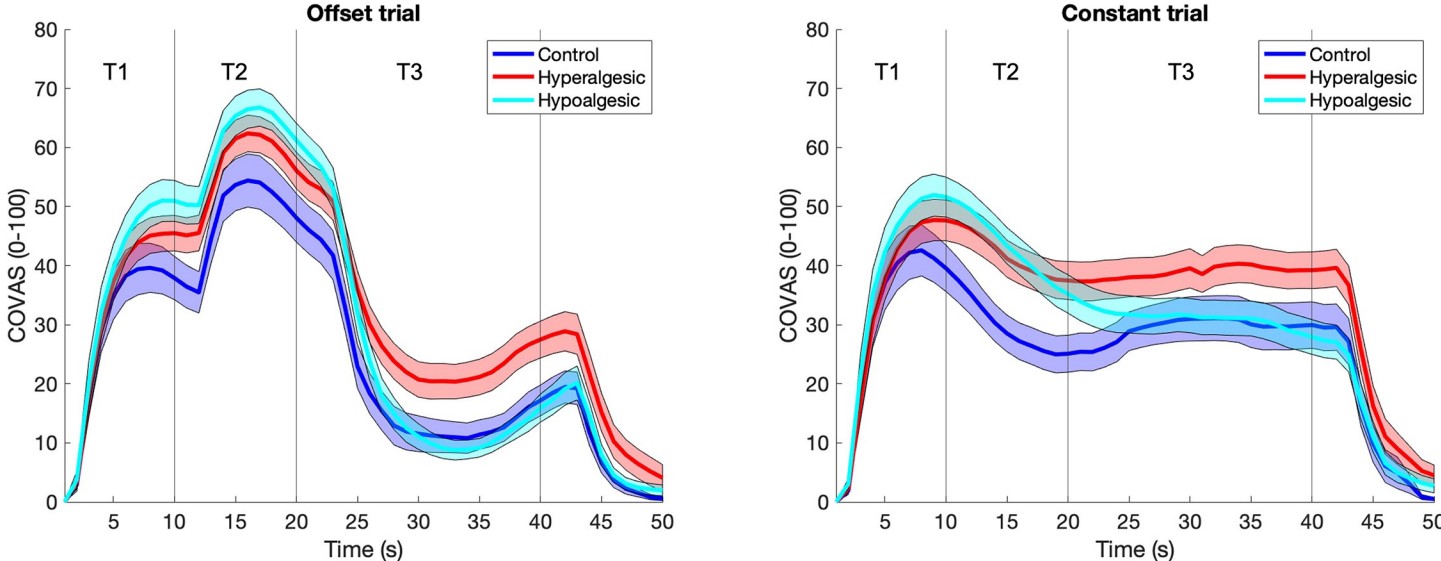

**Fig 2.** Pain ratings in offset analgesia (left) and constant trials (right). Note that in offset analgesia trials, pain was disproportionally reduced during the last 20s of thermal stimulation assessed via a computerized visual analog scale (COVAS). Hyperalgesic suggestion inhibited the development of profound analgesia present in the control as well as the hypoalgesic group. Suggestion affected constant trials in a similar fashion. Bold curves represent mean pain whereas shaded zones are standard errors of the mean (SEM).

During OT, a significant difference for the factor "group" was found between the three groups in pain ratings at T3 ($F_{[2, 94]}$ = 4.81, p = 0.01, $\eta^2_p$ = 0.10). Bonferroni-corrected post-hoc t-tests showed significantly greater pain in the hyperalgesic group than in both the hypoalgesic (p = 0.03) and control (p = 0.02) groups. In contrast, no significant difference was found between the hypoalgesic and the control group (p = 1.00, see comparisons on Fig 3). Results

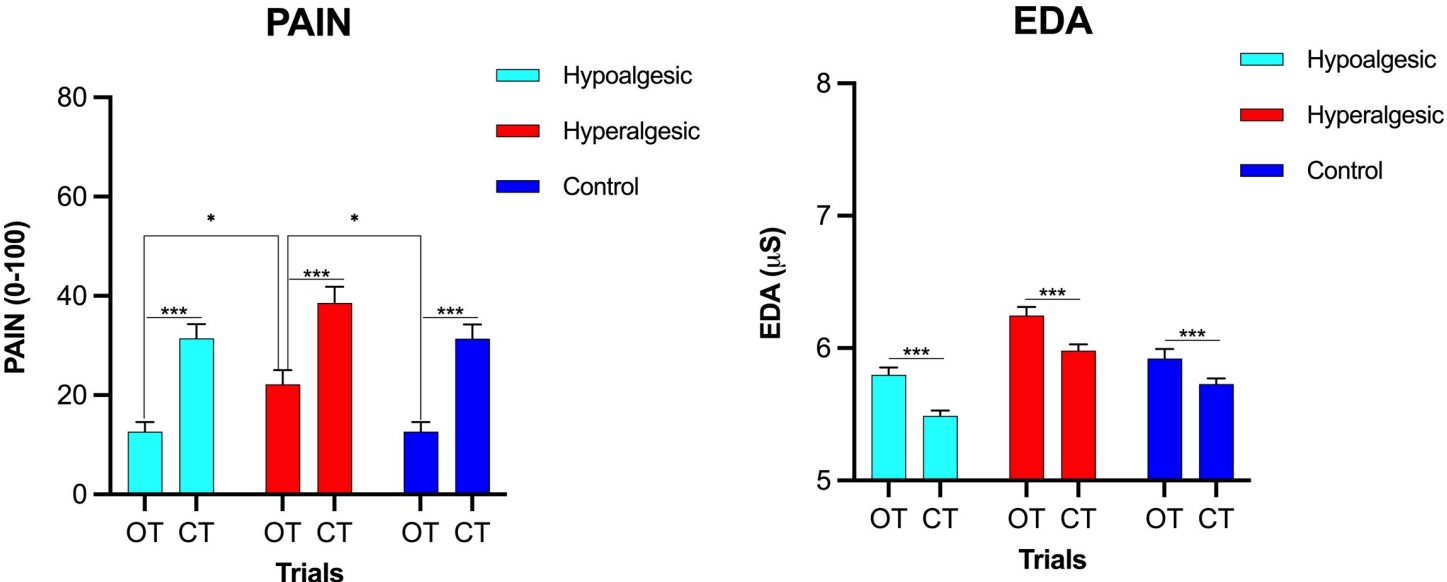

**Fig 3.** Within- and between-group effects for pain assessed via a computerized visual analog scale (COVAS, left) and electrodermal activity (EDA, right). Offset analgesia was reduced in the hyperalgesic group as reflected by a less pronounced difference in pain (averaged of 25-34s interval) between offset trials (OT) and constant trials (CT). Upper comparisons denote between-group comparisons: The hyperalgesic group experienced more pain than the control and hypoalgesic groups. Lower comparisons denote within-group comparisons for OT and CT. Error bars represent standard errors of the mean (SEM), * indicates a significant difference at p < 0.05, *** p < 0.001.

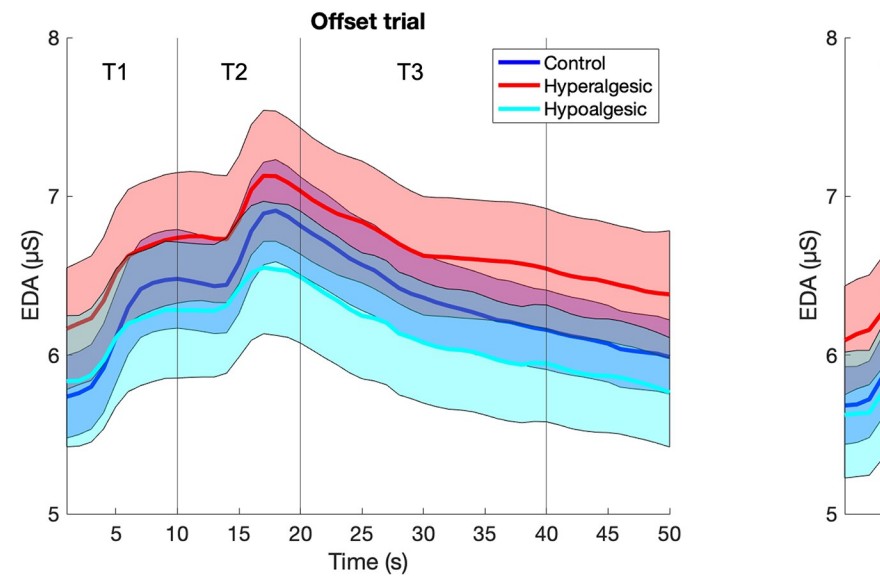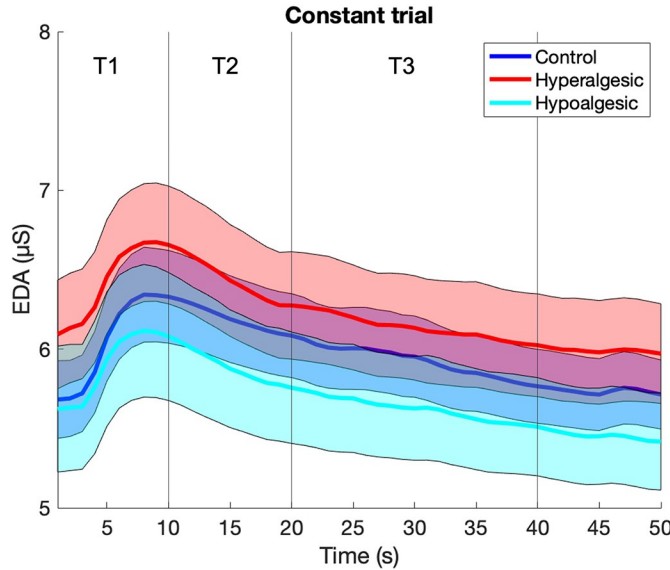

**Fig 4.** Electrodermal activity (EDA) in offset analgesia (left) and constant trials (right). Compared to constant trials, offset analgesia produced paradoxically higher EDA levels during the T3 interval. Bold curves represent mean pain whereas shaded zones are standard errors of the mean (SEM). Vertical markers separate T1, T2 and T3 intervals.

for an additional method of analysis are presented in the supporting information (S2 File). Regarding the CT (T3 interval), no significant difference was shown between all groups ($F_{[2, 94]}$ = 2.08, p = 0.13, $\eta^2_p$ = 0.13), indicating that the verbal suggestion affected OA trials in the hyperalgesic group and not constant trials. Furthermore, no significant difference was shown between the groups regarding EDA in the T3 time interval, neither for OT ($F_{[2, 94]}$ = 0.98, p = 0.38, $\eta^2_p$ = 0.02) nor for CT ($F_{[2, 94]}$ = 0.91, p = 0.40, $\eta^2_p$ = 0.02, Figs 3 and 4).

After completion of the study, 70.1% (n = 68) of the participants stated that they perceived pain during OT that was in line with the provided suggestion. Pain response consistent with the provided suggestion was confirmed by 90.6% (n = 29) of the participants in the hypoalgesic group, but only 21.2% (n = 7) in the hyperalgesic group. A significant difference between groups was observed ($\chi^2_{[1, 65]}$ = 31.7, p < 0.001, Φ = 0.70). 74.2% of participants (n = 72) confirmed this for the CT. Thereby, 75.0% (n = 24) confirmed this in the hypoalgesic, but only 48.5% (n = 16) in the hyperalgesic group ($\chi^2_{[1, 65]}$ = 4.8, p = 0.03, Φ = 0.27).

Secondary analyses of the EDA data within the control group revealed a significant interaction between the factors "interval" (T1, T2, T3) and "trial" (OT, CT) ($F_{[1.4, 62]}$ = 9.0, p = 0.002, $\eta^2_p$ = 0.23). Bonferroni-corrected post-hoc comparisons showed significant differences between OT and CT for the interval T2 (p <0.001) and T3 (p = 0.002), but not for the interval T1 (p = 0.42). Furthermore, within the trial OT significant differences were found between T1 and T2 (p = 0.002) and between T2 and T3 (p < 0.001), but not between T1 and T3 (p = 1.00). A dependent t-test showed that the EDA magnitude of the increase from T1 to T2 did not differ from the magnitude of the reduction from T2 to T3 ($T_{[31]}$ = 0.21, p = 0.84, $d_z$ = 0.04).

No significant correlations were found between the previously described questionnaires and the T3 pain response (OT) in the hypoalgesic group as well as in the control group (p > 0.05, r < 0.3). However, in the hyperalgesic group a significant correlation with the LOT-R was shown (r = -0.45, p < 0.01), which can be attributed mainly to optimism characteristics (optimisms score: r = -0.55, p < 0.01) and not to pessimism characteristics (pessimisms score: r = 0.27 (p = 0.13). All correlation results are presented in the supporting information (S1 Table).

## Discussion

In summary, it can be concluded that OA was provoked in all groups, independent of the suggestion manipulation. However, the pain response but not the EDA response during an OA paradigm was influenced by visually reinforced verbal suggestion in healthy participants via hyperalgesic suggestion, but not via hypoalgesic suggestion.

### Expectancy mechanism

To the best of our knowledge, this is the first study that has attempted to influence OA using suggestion. However, similar results have already been reported for studies that attempted to influence outcomes using other paradigms to quantify endogenous pain modulation by using suggestion. For example, a similar conclusion was reported by Vaegter et al. (2020) which attempted to influence exercise-induced hypoalgesia (EIH) using suggestion [37]. In that study, EIH was defined as an increased pain threshold and pain tolerance induced by performing a single exercise routine. It was found that volunteers who received a negative suggestion prior to exercise, experienced hyperalgesia instead of EIH. Furthermore, studies found that CPM can also be influenced by suggestions and thereby altered expectation [26, 38]. In CPM, the pain response to a painful test stimulus is inhibited by the application of a distant painful conditioning stimulus [39]. Goffaux et al. (2007) studied 20 healthy volunteers regarding their pain perception during the CPM paradigm while they were given different verbal suggestions about the expected pain process [38]. While the hypoalgesic group experienced profound analgesia, this was absent in the hyperalgesic group. Moreover, Bjørkedal and Flaten (2012) also found an effect of verbal suggestion on pain perception in the CPM paradigm [26]. It should be noted, however, that OA is not based on the same mechanisms as CPM and EIH, making them not directly comparable. For example, CPM, unlike OA, can be influenced by ketamine [40], and the two paradigms have underlying distinct brain mechanisms [41]. In addition, other studies have shown that there is no correlation between OA and EIH [42] or OA and CPM [41, 43], also suggesting individual mechanisms of these pain modulation phenotypes. However, based on these similar results for CPM, EIH, and OA, it is reasonable to assume that altered expectations manipulated by suggestion influence endogenous pain modulation processes as quantified via various paradigms.

In general, it can be assumed that suggestion influences OA, since brain activity during OA overlaps with the activity during placebo analgesia [8, 44]. Previous studies have shown that especially the activation of the rostral anterior cingulate cortex (rACC) and the dorsolateral prefrontal cortex (DLPC) play a major role in placebo analgesia [45, 46]. They are both functionally connected with the periaqueductal gray and the rostral ventromedial medulla (PAG-RVM system), which can send inhibitory projections to the spine and thereby elicit a diffuse analgesic response [47]. Increased activation of the DLPC and PAG-RVM circuits were also found during OA [13, 15, 41], suggesting that the mechanisms of placebo analgesia and OA may be similar. However, this is contradicted by the results that hypoalgesic suggestion did not produce increased pain reduction in OA in this study. This can be explained by the fact, that previous studies have shown that placebo analgesia is mediated primarily by increased release of endogenous opiates [48], whereas OA has been shown to be opioid-independent [49]. For example, placebo analgesia can be blocked by the opioid antagonist naloxone [50], whereas naloxone, on the other hand, has no effect on the magnitude of OA [49]. One study has found that nocebo hyperalgesia is mediated by the neurotransmitter cholecystokinin (CCK) [51]. However, the effect of CCK on OA has not yet been studied, although it is relevant since the hyperalgesic manipulation could have influenced the OA.

Interestingly, no significant effect of the suggestion was found on CT in this study. Pain perception in the T3 interval of the CT was neither increased nor decreased. This result is in contrasts with other reports because, in principle, both hypo- or hyperalgesic suggestion have been found in previous studies to influence a wide variety of noxious stimuli [21, 52]. For example, the study by van Laarhoven et al. (2011) found an effect of hyperalgesic verbal suggestion on pain perception in healthy women. However, this study did not use tonic heat stimuli, but mechanical and electrical stimuli [23]. The methodology of other studies also differed in many ways from the present study. For example, studies often used other stimulus modalities (e.g., cold, electric shocks, ischemic pain), or did not use verbal and visual suggestion but used either conditioning alone or a combination of conditioning and suggestion to influence pain perception [21, 52]. Furthermore, no study was found that investigated the effect of verbal and visual suggestion on a constant (tonic) heat stimulus, as done in this study. However, one explanation for the differences in influences on CT versus OT could be the difference in physiological processing. It is suggested that pain adaptation to a moderate, constant heat stimulus is primarily mediated by peripheral mechanisms [53–55]. In comparison, both peripheral and central mechanisms are known to shape OA [8]. Since peripheral mechanisms cannot be influenced by suggestion, this could be a possible explanation for the lack of influence on CT. At the same time, the shown suggestibility during OT could support the assumption that OA is primarily a central phenomenon, as pain perception was modulated by expectancy, here. Further studies comparing the suggestibility of responses to OT and CT are needed to draw further conclusions about the underlying mechanisms.

As a limitation, suggestions might not have been fully successful, as shown by the results of the manipulation check. Although the majority (70.1%) of the subjects reported that they perceived a pain response during OT according to the prior given suggestion. However, in contrast to the hypoalgesic group (90.6%), only 21.2% in the hyperalgesic group confirmed a pain response as suggested. Thus, it can be assumed that one reason for this may be the exceptionally robust analgesia during the OA paradigm. Thus, these results show that OA can be influenced only unidirectionally, being more likely to be enhanced but more difficult to be inhibited.

## Physiological mechanisms

So far, physiological measurements taken during OA included functional magnetic resonance imaging [12, 13, 15, 17, 41], electroencephalography [56] and functional near-infrared spectroscopy [57]. To the best of our knowledge, this is the first experiment which recorded EDA during OA. This was done for the following reasons: Firstly, OA has been shown to be mediated by the activation of brain areas associated with the regulation of autonomic reactivity [13, 15, 58, 59], thus we aimed to capture this variable continuously to test if OA measurements behaviorally overlap with physiological responses. Secondly, we aimed to investigate if verbal suggestion alters both, subjective and objective outcomes during an OA paradigm.

Interestingly, results of this study showed that, contrary to our prediction, EDA responses to OT were not decreased alongside pain perception. In turn, the increase of the temperature during a T2 interval significantly elevated the EDA level which persisted during the T3 interval, whereas the pain response was reduced. Indeed, a lowered pain intensity overlapped with higher EDA level in offset compared to constant trial. It can be suggested that the higher stimulus in T2 of the OT activates the descending pain inhibition pathways and therefore inhibits pain. In fact, the EDA level during CT gradually decreased over-time which was, in general, associated with higher pain compared to OTs. During OA, endogenous modulatory mechanisms have been shown to be activated [9], which are believed to be driven by PAG activation

[13]. PAG is an anatomic structure with multiple nociceptive projections and it plays a crucial role in control of autonomic functions [58, 59]. Whether the enhanced activation in PAG explains reversed offset in EDA needs to be determined.

However, whether this EDA response represents a physiological correlate of the OA via pain response is not clear. Indeed, no differences were found for the EDA data within the OT (as is usual for the pain response) when the time intervals T1 and T3 were considered. One can therefore assume that the increased EDA response within T3, may related to an (still) ongoing EDA response caused by a higher stimulation during the T2 interval. Thus, it should be further investigated whether the EDA within an OA paradigm may also reflect the pain predictions that may involve perception of both temperature decreases (OA) and temperature increases (onset hyperalgesia) as shown by Alter et al. 2020 [60].

Our psychological manipulation did not influence the EDA signal. Although EDA has been used extensively to capture reported pain intensity and has been shown to be a potent biomarker in pain prediction [61], we could not observe that autonomic reactivity was influenced by verbal suggestion. Similar results have been reported in some of the previous experiments [62], in which verbal suggestions towards analgesia or hyperalgesia were provided [63]. It cannot be, however, excluded that this is a result of the relatively small effect size observed at the behavioral level (pain).

### Psychological mechanisms

The results of this study could serve as an explanatory approach to describe why OA is reduced in chronic pain patients. Various studies showed that a large proportion of chronic pain patients have dysfunctional beliefs about their condition and dysfunctional coping strategies in dealing with their condition [64]. It can be hypothesized that because of these dysfunctional beliefs and coping strategies, chronic pain patients have a fundamentally more negative expectancy toward pain. In this study, a negative expectancy was also evoked in the nocebo group by a suggestion in healthy participants. These participants also subsequently showed reduced OA. Thus, the expectation towards pain could have a decisive influence on the magnitude of OA. For the reduced OA in the nocebo group, the optimism of a person seems to play a role. According to the results, it can be assumed that a more pronounced optimism reduces the effect of suggestion of the nocebo group on OA. Thus, the individual could protect from a more pronounced hyperalgesia as a result of the received suggestion. In contrast, no significance of other psychological factors considered for the effect of the suggestions could be observed.

### Conclusion

In this study, suggestion manipulations have been shown to effectively reduce, but not increase, the pain response during OA in healthy participants. Using EDA, this pattern of responses was not observed.

### Supporting information

**S1 Fig. Schematic representation of the heat stimuli and the suggestion figures of the expected pain perception.** Heat stimuli within the Offset Trial (A): T1 interval (0–9 sec) at 46˚C, T2 interval (10–19 sec) at 47˚C, T3 interval (20–40 sec) at 46˚C. Heat stimuli within the Constant Trial (B): constant at 46˚C; suggestion figures of the hypoalgesic group during the Offset Trial (C), pain perception first increases to a level of 50/100, then to 70/100 and drops sharply in the last seconds to an almost non-painful level (approx. 5/100); during the Constant Trial (D), pain perception starts at a level of 50/100 and then slowly and constantly decreases;

suggestion images of the hyperalgesic group: during the Offset Trial (E), pain perception first increases to a level of 50/100, then to 70/100 and finally to 50/100 again; during the Constant Trial (F), pain perception remains constant at a level of 50/100.
(PDF)

**S1 File. Standardized verbal suggestions.**
(DOCX)

**S2 File. Additional method of analysis for offset analgesia.**
(DOCX)

**S3 File. Data set for the analysis.**
(XLSX)

**S1 Table. Correlation analysis of pain scores within the third time interval (T3) and included questionnaires.** PHQ9: Patient Health Questionnaire; PVAQ: Pain Vigilance and Awareness Questionnaire; PSQ: Pain Sensitivity Questionnaire; STAIT-SKD: State-Trait-Anxiety-Inventory-SKD; SDS-17: Social Desirability Scale-17; MAAS: Mindful Attention and Awareness Scale, LOT-R: Life-Orientation-Test, r: spearman-correlation coefficient, p: p-value, significant correlations are marked in bold.
(DOCX)

## Acknowledgments

The authors thank the Institute of Medical Informatics, University of Lübeck, kindly for providing the research facilities and equipment.

## Author Contributions

**Conceptualization:** Tibor M. Szikszay, Waclaw M. Adamczyk, Kerstin Luedtke.

**Data curation:** Tibor M. Szikszay, Philip Gouverneur.

**Formal analysis:** Tibor M. Szikszay, Philip Gouverneur.

**Funding acquisition:** Kerstin Luedtke.

**Investigation:** Janina Panskus, Carolin David.

**Methodology:** Tibor M. Szikszay, Waclaw M. Adamczyk, Janina Panskus, Lotte Heimes, Carolin David.

**Project administration:** Janina Panskus, Lotte Heimes, Carolin David.

**Supervision:** Tibor M. Szikszay, Waclaw M. Adamczyk, Kerstin Luedtke.

**Validation:** Tibor M. Szikszay.

**Writing – original draft:** Tibor M. Szikszay, Waclaw M. Adamczyk.

**Writing – review & editing:** Janina Panskus, Lotte Heimes, Carolin David, Philip Gouverneur, Kerstin Luedtke.

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
