## [Decision Letter · Decision Letter 0]

9 Nov 2022

PONE-D-22-22989Psychological mechanisms of offset analgesia The effect of expectancy manipulationPLOS ONE

Dear Dr. Szikszay,

Thank you for submitting your manuscript to PLOS ONE. After careful consideration, we feel that it has merit but does not fully meet PLOS ONE’s publication criteria as it currently stands. Therefore, we invite you to submit a revised version of the manuscript that addresses the points raised during the review process.

Please address the concerns raised by both reviewers, particularly the major issues pointed out by Reviewer 1.

We look forward to receiving your revised manuscript.

Kind regards,

Kelly Naugle, PhD

Academic Editor

PLOS ONE

Journal Requirements:

 “KL - study conducted thanks to support from the National Science Centre in Poland (2020/04/X/HS6/01927)”

Reviewers' comments:

Reviewer's Responses to Questions

**Comments to the Author**

1. Is the manuscript technically sound, and do the data support the conclusions?

Reviewer #1: Partly

Reviewer #2: Yes

2. Has the statistical analysis been performed appropriately and rigorously? 

Reviewer #1: Yes

Reviewer #2: Yes

3. Have the authors made all data underlying the findings in their manuscript fully available?

Reviewer #1: No

Reviewer #2: Yes

4. Is the manuscript presented in an intelligible fashion and written in standard English?

Reviewer #1: Yes

Reviewer #2: Yes

5. Review Comments to the Author

Reviewer #1: “Psychological mechanisms of offset analgesia The effect of expectancy manipulation” is a well-written article describing a fascinating experiment designed to expand the understanding of offset analgesia, which is likely to involve descending inhibition of nociceptive input. Hypothesizing that offset analgesia shares mechanisms with placebo analgesia, they reasoned that one well-known component of placebo analgesia, hypoalgesic suggestion, might amplify offset analgesia. Given the bidirectional nature of descending pain modulation, the investigators also reasoned that hyperalgesic suggestion might attenuate offset analgesia. The study was well executed in a large sample of healthy volunteers. With the currently presented analysis, the hypoalgesic suggestion did not increase offset analgesia compared to control, but instead, a hyperalgesic suggestion decreased offset analgesia. With major changes, this investigation would be a nice addition to the field.

Major issues:

• Data availability – the authors need to specify where the data are and/or provide it for review.

• Reporting pain intensity change as a measure of offset analgesia can be done in several ways, as outlined by the current authors elsewhere (Szikszay et al. Clin J Pain 2018). Statistical testing comparing OT and CT is described in the current analysis. However, within-stimulus subtraction should also be analyzed to determine whether similar effects of suggestion are observed. This is particularly interesting because, from inspecting Fig 2, it seems that if one were to use the within-stimulus measure of offset analgesia used by Niesters et al Anesthesiology 2011 and others ([min – max] / max), the hypoalgesic suggestion may be associated with relatively greater offset analgesia, which would support the investigators’ a priori hypothesis.

• Please provide additional information about the EDA analysis. Is there any signal processing done by the acquiring unit? Is there any preprocessing besides downsampling from 1000 Hz to 1 Hz? How was the downsampling done (program used, method)?

• More should be done in the analysis of the EDA signals. In the control group (N=30), a secondary analysis should be done to determine if there is a significant difference between control and offset stimuli at all timepoints (time series data) and then at timepoints during the T1, T2, and T3 periods. The temporal relationship of the EDA trace to the temperature stimulus should also be highlighted, potentially with markers of the temperature transitions. This analysis would contribute to the field, since this is the first EDA measure during offset analgesia and would support statements in the Discussion about whether EDA reflects a physiological correlate of offset analgesia

• The second increase in EDA (potentially reflecting the transition from 47 to 46 C) may be consistent with offset analgesia when considering that offset analgesia may reflect pain predictions, which themselves may elicit pain increases or decreases during the offset stimulus. As part of the above analysis, it would be interesting to see whether the magnitude of the EDA increase is different with the temperature increase versus decrease (i.e. comparing EDA during T2-T1 versus T3-T2). Along similar lines, the initial increase in EDA may actually relate to a change in pain and not pain intensity per se. Perceptual enhancement of both temperature decreases (offset analgesia) and increases (onset hyperalgesia) has been observed using similar heat stimuli (Alter et al. PLOSOne 2020), so the EDA response may actually be related to pain prediction.

• Is the manipulation placebo or nocebo? There is no administration of an inert compound. The relationship of the described suggestion manipulation to placebo / nocebo effects is highly relevant and interesting, but it would be better to refer to the current manipulation as hypo- or hyperalgesic suggestion, as done in the abstract. Please revise other sections accordingly.

Minor issues:

• Would add a summary paragraph at the end of the Discussion

• Fig 3 – it is unclear which specific comparisons are significant. This is particularly true in the left panel, where it’s hard to tell what the comparison is for the single stars – is this comparing hypo vs hyper and hyper vs control or the OT for each of those conditions? A bracket with descending lines pointing to the specific bars or bar groups might be considered.

• Pg 8 line 156 – would rephrase or move paragraph to the section about the cover story. Rephrasing by stating the purpose of the cover story first would help the flow for the reader.

• P 14 line 275-277 – please rephrase. Ambiguous what “this” is.

• P 15 line 288 – not exclusively a “verbal” manipulation – please rephrase

• P 16 line 310-311 – rephrase last phrase in sentence. Perhaps, “have underlying” instead of “are underlying”

• P 16 line 325 – rephrase “placebo suggestion.” Would delete placebo and stick with terminology in earlier sections.

• P17 line 360 – please rephrase to clarify the different percentages here.

Reviewer #2: I have no concerns about this research. I feel like the manuscript was very well written and only needs minor corrections before publication. The manuscript is also appropriate for the scope and aim of PlosOne.

6. PLOS authors have the option to publish the peer review history of their article (what does this mean?). If published, this will include your full peer review and any attached files.

Reviewer #1: **Yes: **Benedict Alter

Reviewer #2: No

---

## [Author Response · Author response to Decision Letter 0]

13 Dec 2022

Reviewer #1: 

“Psychological mechanisms of offset analgesia The effect of expectancy manipulation” is a well-written article describing a fascinating experiment designed to expand the understanding of offset analgesia, which is likely to involve descending inhibition of nociceptive input. Hypothesizing that offset analgesia shares mechanisms with placebo analgesia, they reasoned that one well-known component of placebo analgesia, hypoalgesic suggestion, might amplify offset analgesia. Given the bidirectional nature of descending pain modulation, the investigators also reasoned that hyperalgesic suggestion might attenuate offset analgesia. The study was well executed in a large sample of healthy volunteers. With the currently presented analysis, the hypoalgesic suggestion did not increase offset analgesia compared to control, but instead, a hyperalgesic suggestion decreased offset analgesia. With major changes, this investigation would be a nice addition to the field.

Response: We would like to thank the reviewer for the valuable time to review our manuscript. We are very pleased with the positive and constructive feedback. We also believe that the proposed and implemented changes have significantly improved the quality of our manuscript.

Major issues:

• Data availability – the authors need to specify where the data are and/or provide it for review.

Response: Thank you for your comment. We would like to attach our dataset as supporting information (please see S4 File). In this way, data will be available without any restrictions.

“The raw data are presented in the supporting information (S4 File).” (p. 11, line 231)

• Reporting pain intensity change as a measure of offset analgesia can be done in several ways, as outlined by the current authors elsewhere (Szikszay et al. Clin J Pain 2018). Statistical testing comparing OT and CT is described in the current analysis. However, within-stimulus subtraction should also be analyzed to determine whether similar effects of suggestion are observed. This is particularly interesting because, from inspecting Fig 2, it seems that if one were to use the within-stimulus measure of offset analgesia used by Niesters et al Anesthesiology 2011 and others ([min – max] / max), the hypoalgesic suggestion may be associated with relatively greater offset analgesia, which would support the investigators’ a priori hypothesis.

Response: Thank you very much for this comment. We performed the statistical analyses based on the T3 interval of OT and CT, as described in our pre-registration. However, we also agree that an analysis with parameters within the OT are valid. We would therefore like to present the suggested ([min - max] / max) analysis carried out by Niesters et al. (2011), in the supporting information (see S3 File). Interestingly, this analysis does not change the overall conclusions derived from the current study but confirms again that participants in the hyperalgesic group perceived a significantly different offset analgesia effect than the hypoalgesic group and the control group. This strengthens our previous findings. We would like to thank you for this suggestion.

“An additional method of analysis calculating the percentage difference of the maximum pain ratings for T2 and the minimum pain rating for T3 is included in the supporting information (S3 File).” (p. 10, line 213, ff)

“Results for an additional method of analysis are presented in the supporting information (S3 File).” (p. 14, line 276, ff)

• Please provide additional information about the EDA analysis. Is there any signal processing done by the acquiring unit? Is there any preprocessing besides downsampling from 1000 Hz to 1 Hz? How was the downsampling done (program used, method)?

Response: Thank you for this comment. We have now included additional information on the EDA analysis in our manuscript. EDA data were measured with a sampling rate of 1000 Hz (PLUX Wireless Biosignals, S.A., Portugal) (please see p. 6, line 125). Downsampling of the time-series data was performed by using the ‘resample’ method of the ‘pandas’ package in Python (Python 3.9.7, pandas 1.4.2). There were no further preprocessing steps.

“The time-series data were down-sampled to a frequency of 1 Hz by using the “resample” function of the python package “Pandas” (Python 3.9.7, pandas 1.4.2). Here, multiple data points are aggregated and replaced by their average. No further preprocessing steps were performed.” (p. 9, line 190, ff)

• More should be done in the analysis of the EDA signals. In the control group (N=30), a secondary analysis should be done to determine if there is a significant difference between control and offset stimuli at all timepoints (time series data) and then at timepoints during the T1, T2, and T3 periods. The temporal relationship of the EDA trace to the temperature stimulus should also be highlighted, potentially with markers of the temperature transitions. This analysis would contribute to the field, since this is the first EDA measure during offset analgesia and would support statements in the Discussion about whether EDA reflects a physiological correlate of offset analgesia

Response: Thank you very much for this comment. We would like to provide further analyses. However, we would like to propose analyses based not on all time points (time series, each second), but on defined time intervals. For this purpose, we would like to suggest the pain ratings and EDA data of the last 5 seconds of the T1 and T2 interval and the mid 10 seconds of the T3 interval (please see p. 5, line 199 following). As shown in Fig. 2 and 4, this corresponds to a constant value in time for pain and EDA response, respectively, in which the signal does not increase or decrease continuously. Likewise, on the one hand, the extracted time intervals (T1, T2, T3), but also the performed statistical models (general linear model) are thus comparable with other analyses here in the manuscript (as also described in the preregistration). As you suggested, in a secondary analysis, we now analyzed differences between OT and CT of EDA response for T1, T2, and T3. Likewise, we analyzed differences between T1, T2, and T3 within OT for the EDA response. No significant differences were found for T1 between OT and CT, but significant differences were found between T2 and T3. However, no significant differences were found within the OT trial between T1 and T3 but between T1 and T2 and between T2 and T3. We now describe these in the results section (p. 15, line 297 following) and added it to the discussion (p. 20, line 413 following). Furthermore, we have highlighted temperature intervals in Figures 2 and 4.

“Secondary analyses of the EDA data within the control group revealed a significant interaction between the factors “interval” (T1, T2, T3) and “trial” (OT, CT) (F[1.4, 62] = 9.0, p = 0.002, �2p = 0.23). Bonferroni-corrected post-hoc comparisons showed significant differences between OT and CT for the interval T2 (p <0.001) and T3 (p = 0.002), but not for the interval T1 (p = 0.42). Furthermore, within the trial OT significant differences were found between T1 and T2 (p = 0.002) and between T2 and T3 (p < 0.001), but not between T1 and T3 (p = 1.00).” (p. 15, line 297 ff)

• The second increase in EDA (potentially reflecting the transition from 47 to 46 C) may be consistent with offset analgesia when considering that offset analgesia may reflect pain predictions, which themselves may elicit pain increases or decreases during the offset stimulus. As part of the above analysis, it would be interesting to see whether the magnitude of the EDA increase is different with the temperature increase versus decrease (i.e. comparing EDA during T2-T1 versus T3-T2). Along similar lines, the initial increase in EDA may actually relate to a change in pain and not pain intensity per se. Perceptual enhancement of both temperature decreases (offset analgesia) and increases (onset hyperalgesia) has been observed using similar heat stimuli (Alter et al. PLOSOne 2020), so the EDA response may actually be related to pain prediction.

Response: Thank you for this comment. We have performed the suggested analysis of the EDA data. The difference between T2 and T1 was not significantly different from the difference between T3 and T2 (p = 0.84). In contrast to the increased EDA signal in T3 of the OT compared to the CT, the lack of difference when T1 and T3 time intervals were considered does not indicate a physiological correlate. We would like to consider this in the discussion. We would also like to point out that for a definitive statement on this, this should be further investigated. For this purpose, the onset hyperalgesia paradigm suggested by you is particularly well suited.

 “A dependent t-test showed that the EDA magnitude of the increase from T1 to T2 did not differ from the magnitude of the reduction from T2 to T3 (T[31] = 0.21, p = 0.84, dz = 0.04).” (p. 15, line 303 ff)

“However, whether this EDA response represents a physiological correlate of the OA via pain response is not clear. Indeed, no differences were found for the EDA data within the OT (as is usual for the pain response) when the time intervals T1 and T3 were considered. One can therefore assume that the increased EDA response within T3, may related to an (still) ongoing EDA response caused by a higher stimulation during the T2 interval. Thus, it should be further investigated whether the EDA within an OA paradigm may also reflect the pain predictions that may involve perception of both temperature decreases (OA) and temperature increases (onset hyperalgesia) as shown by Alter et al. 2020 [61].”(p. 20, line 413 ff).

• Is the manipulation placebo or nocebo? There is no administration of an inert compound. The relationship of the described suggestion manipulation to placebo / nocebo effects is highly relevant and interesting, but it would be better to refer to the current manipulation as hypo- or hyperalgesic suggestion, as done in the abstract. Please revise other sections accordingly.

Response: Thank you for this comment. The reviewer is right that no administration of a physical treatment serving as a vehicle for placebo effect seems odd. However, we would like point to the fact that contemporary understanding of placebo effects, i.e., placebo analgesia (hypoalgesia) and nocebo hyperalgesia (in the context of pain) exceeds far beyond the mere administration of the inert treatment. According to Benedetti et al. (2011), a context makes the given procedural maneuvers a placebo (or nocebo) leading to placebo (or nocebo) effects. Furthermore, in the seminal work by Colloca & Miller (2011), authors introduced Peirce’s theory of sign, wherein the mere information seems to be a core to trigger expectations leading eventually to behavioral change (e.g., the placebo effect). Thus, any sort of procedure seems to convey an information, and this is its meaning to patients which provokes the placebo effect. In our study, the information was explicitly verbalized and had a form of verbal suggestion. Furthermore, the reviewer is right that the terms should be used more consistently. We have revised the manuscript and used the terms hypo- or hyperalgesic suggestion consistently.

References:

• Benedetti F. (2006). Placebo analgesia. Neurological sciences: official journal of the Italian Neurological Society and of the Italian Society of Clinical Neurophysiology, 27 Suppl 2, S100–S102. https://doi.org/10.1007/s10072-006-0580-4

• Colloca, L., & Miller, F. G. (2011). How placebo responses are formed: a learning perspective. Philosophical transactions of the Royal Society of London. Series B, Biological sciences, 366(1572), 1859–1869. https://doi.org/10.1098/rstb.2010.0398

Minor issues:

• Would add a summary paragraph at the end of the Discussion

Response: We have included a summary at the end of the discussion:

“In this study, suggestion manipulations have been shown to effectively reduce, but not increase, the pain response during OA in healthy participants. Using EDA, this pattern of responses was not observed.” (p. 21, line 446 ff)

• Fig 3 – it is unclear which specific comparisons are significant. This is particularly true in the left panel, where it’s hard to tell what the comparison is for the single stars – is this comparing hypo vs hyper and hyper vs control or the OT for each of those conditions? A bracket with descending lines pointing to the specific bars or bar groups might be considered.

Response: You are correct, the original figure with marked comparisons was not straightforward. We have now revised Fig 3. The upper comparisons denote comparisons between the groups of OTs. The lower comparisons indicate differences between OTs and CTs from T3 within each group.

• Pg 8 line 156 – would rephrase or move paragraph to the section about the cover story. Rephrasing by stating the purpose of the cover story first would help the flow for the reader.

Response: Thank you very much for this comment. We would like to describe the cover story prior to the suggestions.

• P 14 line 275-277 – please rephrase. Ambiguous what “this” is.

Response: We have now clarified this:

„Pain response consistent with the provided suggestion (…).” (p. 14, line 291-292)

• P 15 line 288 – not exclusively a “verbal” manipulation – please rephrase

Response: We have adapted this as well.

• P 16 line 310-311 – rephrase last phrase in sentence. Perhaps, “have underlying” instead of “are underlying”

Response: Thank you very much, we have corrected it.

• P 16 line 325 – rephrase “placebo suggestion.” Would delete placebo and stick with terminology in earlier sections.

Response: We have corrected this as well.

• P17 line 360 – please rephrase to clarify the different percentages here.

Response: We have adapted this as follows:

“Although the majority (70.1%) of the subjects reported that they perceived a pain response during OT according to the prior given suggestion. However, in contrast to the hypoalgesic group (90.6%), only 21.2% in the hyperalgesic group confirmed a pain response as suggested.” (p. 18, line 387 ff)

Reviewer #2: 

I have no concerns about this research. I feel like the manuscript was very well written and only needs minor corrections before publication. The manuscript is also appropriate for the scope and aim of PlosOne.

Response: Thank you for taking the time to review the manuscript. We are very happy about your positive feedback.

Additional comments

General: The current manuscript looks to examine the psychological mechanisms related to offset analgesia. While many mechanisms have been examined in relation to offset analgesia, the use of psychological interventions is a novel methods determining endogenous pain modulation. This study used suggestions to determine the degree of offset analgesia during two separate trials using an experimental heat stimulus. The study found that offset analgesia was present in all groups, regardless of the type of suggestion provided. While I feel the manuscript is very well written, there are some things that can be added to improve the overall quality.

Response: Once again, thank you for taking the time to review the manuscript. We also appreciate your positive feedback.

COMMENTS

Introduction

Lines 64-71: you introduce the terms placebo and nocebo. You should define these terms for the reader. While placebo is fairly common, nocebo is less so and readers may not understand its intent.

Response: Thank you for this comment. As suggested by Reviewer 1, we would like to avoid using the terms "placebo" and "nocebo" as they can be misleading in this context. Therefore, we have replaced these terms with hypo- or hyperalgesic suggestion in the manuscript.

Methods

Lines 98-103: It would help if you included the suggestions used in this section.

Response: Thank you very much for this comment. We have now clarified this:

“…i) the hypoalgesic group with suggestion towards profound hypoalgesia following the temperature reduction, ii) the hyperalgesic group with verbal suggestion towards hyperalgesia following the temperature reduction …” (p. 5, line 100 ff)

Lines 107-108: How did participants confirm they were otherwise healthy? Did you use a PARQ or some other questionnaire?

Response: Participants were asked whether they would subjectively describe themselves as healthy at this point in time. This was confirmed in writing and verbally by all included subjects. We would like to make this clearer as well:

“All participants had to subjectively confirm that they were healthy”. (p. 6, line 108)

Lines 142-144: Explain in more detail how participants rated their perceived pain. Did they use a touchscreen or did they have a mouse? How often were ratings given?

Response: Thank you for this comment. A computerized visual analogue scale device (COVAS, Medoc, Ramat Yishai, Israel) was used for pain assessment. The COVAS is an additional device which is compatible with the Pathway CHEPS thermal stimulator. The COVAS itself transforms the movement of the slider into digital metric of reported pain. Participants are asked to provide a continuous VAS assessment in real time using this slider. All participants received two offset and two constant trials and were asked to rate pain during each trial. We have now clarified this in the following section:

„A computerized visual analog scale hardware device (COVAS; with the range 0 = "no pain" to 100 = "most tolerable pain") was used for continuous assessment of pain intensity (Medoc, Ramat Yishai, Israel).” (p. 6, line 122 ff)

Results

Line 230: Include a closed parathesis after 0.06 and include a space between that and ‘or’

Response: We apologize for this typo. It has been corrected.

---

## [Decision Letter · Decision Letter 1]

2 Jan 2023

PONE-D-22-22989R1Psychological mechanisms of offset analgesia: The effect of expectancy manipulationPLOS ONE

Dear Dr. Szikszay,

Thank you for submitting your manuscript to PLOS ONE. After careful consideration, we feel that it has merit but does not fully meet PLOS ONE’s publication criteria as it currently stands. Therefore, we invite you to submit a revised version of the manuscript that addresses the points raised during the review process.

We look forward to receiving your revised manuscript.

Kind regards,

Kelly Naugle, PhD

Academic Editor

PLOS ONE

Journal Requirements:

Additional Editor Comments:

The manuscript has substantially improved with the revisions and only is in need of minor revisions.  Specifically, as pointed out by Reviewer 2, it appears that the dataset shared does not include the complete dataset.  Also, the timeseries data plotted in Figures 2 and 4 does not appear in the dataset file. Please include a data dictionary for all of the variables. 

Reviewers' comments:

Reviewer's Responses to Questions

**Comments to the Author**

1. If the authors have adequately addressed your comments raised in a previous round of review and you feel that this manuscript is now acceptable for publication, you may indicate that here to bypass the “Comments to the Author” section, enter your conflict of interest statement in the “Confidential to Editor” section, and submit your "Accept" recommendation.

Reviewer #1: (No Response)

Reviewer #2: All comments have been addressed

2. Is the manuscript technically sound, and do the data support the conclusions?

Reviewer #1: Yes

Reviewer #2: Yes

3. Has the statistical analysis been performed appropriately and rigorously? 

Reviewer #1: Yes

Reviewer #2: Yes

4. Have the authors made all data underlying the findings in their manuscript fully available?

Reviewer #1: No

Reviewer #2: Yes

5. Is the manuscript presented in an intelligible fashion and written in standard English?

Reviewer #1: Yes

Reviewer #2: Yes

6. Review Comments to the Author

Reviewer #1: The manuscript is substantially improved. Additionally, the letter response provided some excellent discourse which was much appreciated.

The one issue remaining is that the dataset shared does not include the complete dataset. The timeseries data plotted in Fig 2 and Fig 4 is not in the dataset file. Additionally, a data dictionary should be included for all variables. Prior to publication, please confirm supplemental files have the correct number within the file.

Reviewer #2: (No Response)

7. PLOS authors have the option to publish the peer review history of their article (what does this mean?). If published, this will include your full peer review and any attached files.

Reviewer #1: **Yes: **Benedict J Alter

Reviewer #2: No

---

## [Author Response · Author response to Decision Letter 1]

3 Jan 2023

Journal Requirements:

Response: Thank you for this comment. All cited papers were checked using The Retraction Watch Database. Minor errors were noticed following verification of the accuracy of the reference list. These have been corrected. All changes are marked in yellow.

Additional Editor Comments:

The manuscript has substantially improved with the revisions and only is in need of minor revisions. Specifically, as pointed out by Reviewer 2, it appears that the dataset shared does not include the complete dataset. Also, the timeseries data plotted in Figures 2 and 4 does not appear in the dataset file. Please include a data dictionary for all of the variables. 

Response: Also thank you for this comment. The dataset has been revised. On the first page of the excel table (S4_File) you now find a clear data dictionary. The characteristics follow on the second page. This is followed by the pain ratings and EDA responses divided by the three groups. We believe that the dataset is now clear and concise.

Reviewer #1: 

The manuscript is substantially improved. Additionally, the letter response provided some excellent discourse which was much appreciated.

Response: Thank you again for reviewing our manuscript. We are sure that your suggestions have improved the quality a tremendous amount.

The one issue remaining is that the dataset shared does not include the complete dataset. The timeseries data plotted in Fig 2 and Fig 4 is not in the dataset file. Additionally, a data dictionary should be included for all variables. Prior to publication, please confirm supplemental files have the correct number within the file.

Response: We have revised the dataset. The first page of the Excel spreadsheet (S4_File) now includes a straightforward data dictionary. Following on the second page are the characteristics of the study subjects. Next are the pain ratings and the EDA responses, divided by the three groups (hypoalgesic, hyperalgesic, control). We believe the data set is now clear and concise. We have also corrected the typo in the numbering of the supplemental files.

---

## [Editor Report · Decision Letter 2]

4 Jan 2023

Psychological mechanisms of offset analgesia: The effect of expectancy manipulation

PONE-D-22-22989R2

Dear Dr. Szikszay,

We’re pleased to inform you that your manuscript has been judged scientifically suitable for publication and will be formally accepted for publication once it meets all outstanding technical requirements.

Kind regards,

Kelly Naugle, PhD

Academic Editor

PLOS ONE
---

## [Editor Report · Acceptance letter]

5 Jan 2023

PONE-D-22-22989R2 

Psychological mechanisms of offset analgesia:
The effect of expectancy manipulation 

Dear Dr. Szikszay:

I'm pleased to inform you that your manuscript has been deemed suitable for publication in PLOS ONE. Congratulations! Your manuscript is now with our production department. 

Kind regards, 

on behalf of

Dr. Kelly Naugle 

Academic Editor

PLOS ONE